*Perspective*

# Signalomics for molecular tumor boards and precision oncology of breast and gynecological cancers

Tatiana V Denisenko[1,6], Anna E Ivanova[1,6], Alexey Koval [2], Denis N Silachev[1,2,3], Lee Jia[4], Gennadiy T Sukhikh[1] & Vladimir L Katanaev [2,5✉]

## Abstract

Precision oncology led to the establishment and widespread application of molecular tumor boards (MTBs)—multidisciplinary units combining molecular and clinical assessment of individual cancer cases for swift selection of personalized treatments. Whole-exome or gene panel sequencing, combined with transcriptomic, immunohistochemical, and other molecular analyses, often permits dissection of molecular drivers of a tumor and identification of its potential targetable vulnerabilities, instructing clinical oncologists on sometimes unconventional treatment options. However, cancer drivers are often unleashed mutation-independently, especially in breast and gynecological cancers, and deleterious mutations are not always pathogenic. To complement the MTB arsenal, we chart here the molecular toolset we call Signalomics that permits fast and robust assessment of a panel of oncogenic signaling pathways in fresh tumor samples. Using transcriptional reporters introduced in primary tumor cells, this approach identifies the pathways overactivated in a given tumor and validates their sensitivity to targeted therapies, providing actionable insights for personalized treatment strategies. Integration of Signalomics into MTB workflows bridges the gap between molecular profiling and functional pathway analysis, refining clinical treatment decisions and advancing precision oncology.

**Keywords** Molecular Tumor Board; Oncogenic Signaling Pathways; Signalomics; Precision Oncology; Breast and Gynecological Cancers

Breast cancer (BC) and gynecological cancers, including ovarian (OC), cervical (CC), and endometrial cancer (EC), are among the leading mortality and morbidity cases in women and represent significant medical, social, and economic challenges (Bray et al, 2024; Kerr et al, 2021). The 5-year overall survival rate for OC patients remains ~47%, with over 70% of cases being diagnosed at advanced stages. CC ranks fourth among the most common cancers in women globally, posing a significant health burden, especially in developing countries. EC, primarily affecting postmenopausal women, is the sixth most common female cancer. Advanced and recurrent gynecological cancers are associated with poor prognosis and lack of effective treatments (Wang et al, 2020). BC, on the other hand, holds the highest incidence and mortality rates among all cancer types affecting women, and is the most frequent type of cancer in general (who.int/initiatives/global-breast-cancer-initiative).

Standard treatment approaches for breast and gynecological cancers involve a combination of surgery, chemotherapy, and radiation therapy, primarily aimed at suppressing tumor growth without specifically targeting the molecular drivers of the disease (Nie et al, 2020). However, the non-specific nature of cell death caused by these treatments can lead to harmful side effects. Insufficiency of available targeted therapeutic strategies has contributed to the high mortality rates in patients with these cancers (Heintz et al, 2006). In recent years, the development of molecular diagnostics utilizing next-generation sequencing (NGS) has allowed comprehensive tumor profiling to enable personalized medicine approaches, engaging the power of targeted therapies that directly address the molecular drivers of cancer (Deans et al, 2017; Weberpals et al, 2011).

Different targeted therapies are available for BC, depending on the tumor subtype: estrogen receptor-positive (ER+), human epidermal growth factor receptor 2-positive (HER2+), and triple-negative breast cancer (TNBC) (Polyak and Metzger Filho, 2012). Thus, ER+ tumors are commonly treated with anti-hormonal therapy to suppress estrogen signaling, often in combination with chemotherapy (Lumachi et al, 2015). HER2+ tumors are treated with monoclonal antibodies against HER2 (such as trastuzumab) and small-molecule tyrosine kinase inhibitors (Stanowicka-Grada and Senkus, 2023). There is still no well-established targeted therapy for TNBC and gynecological cancers, despite some recent developments (Bardia et al, 2021; Schmid et al, 2024; Tutt et al, 2021).

Gynecological and breast tumors have lower rates of actionable mutations compared to other solid tumors (Gunderson et al, 2016). Ongoing studies explore

[1]Kulakov National Medical Research Center of Obstetrics, Gynecology and Perinatology, 4 Akademika Oparina Str., Moscow 117997, Russia. [2]Translational Research Centre in Oncohaematology, Department of Cell Physiology and Metabolism, Faculty of Medicine, University of Geneva, CH-1211 Geneva, Switzerland. [3]Department of Functional Biochemistry of Biopolymers, A.N. Belozersky Research Institute of Physico-Chemical Biology, Moscow State University, 119992 Moscow, Russia. [4]College of Materials and Chemical Engineering, Minjiang University, Fuzhou, Fujian 350108, China. [5]Translational Oncology Research Center, Qatar Biomedical Research Institute (QBRI), College of Health and Life Sciences, Hamad Bin Khalifa University (HBKU), Qatar Foundation, PO Box 34110, Doha, Qatar. [6]These authors contributed equally: Tatiana V Denisenko, Anna E Ivanova. ✉E-mail: vkatanaev@hbku.edu.qa
https://doi.org/10.1038/s44320-025-00125-1 | Published online: 9 June 2025

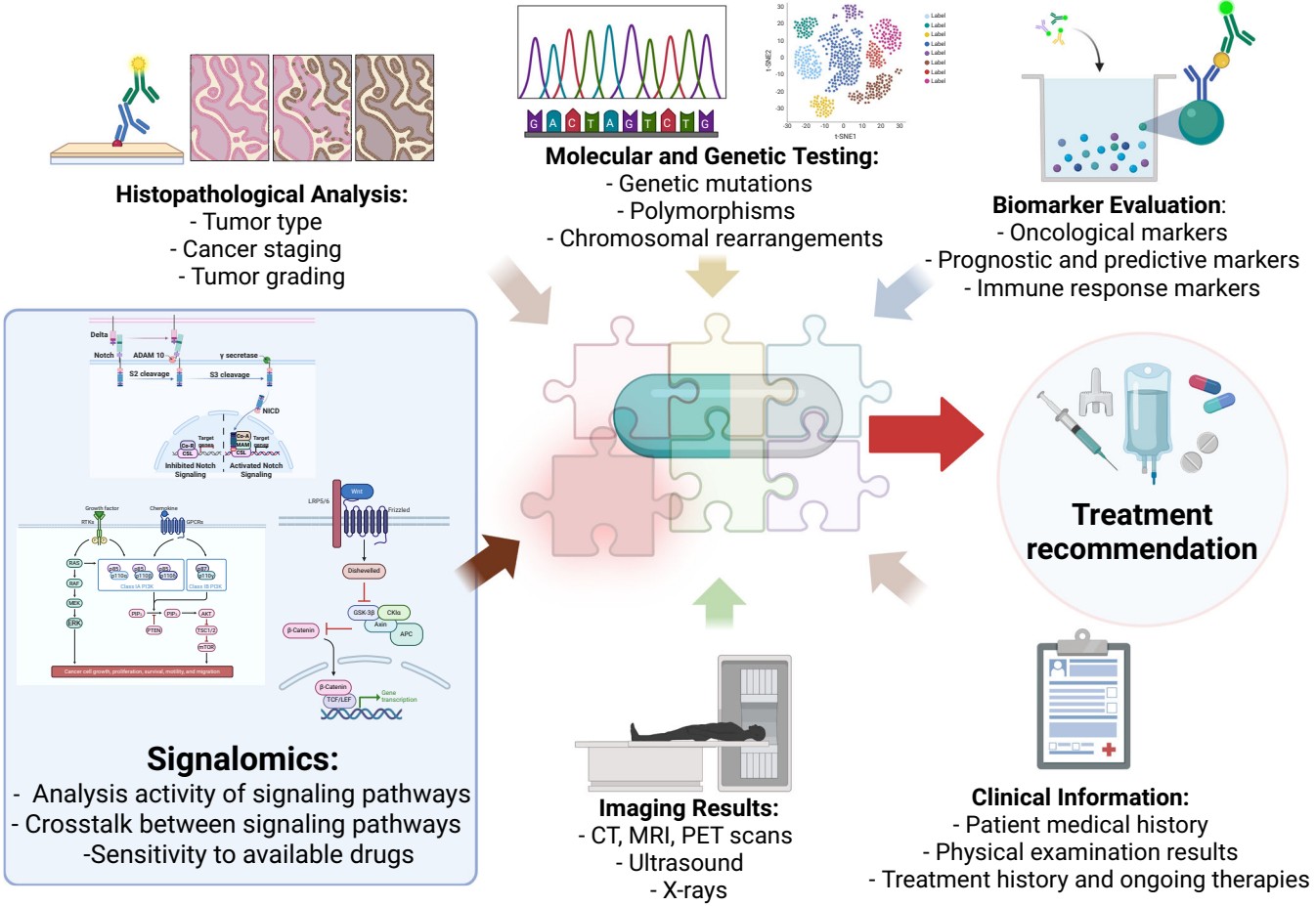

**Figure 1. Signalomics as the missing piece of the puzzle for molecular tumor boards (MTBs).**

Currently, several technological packages (NGS, IHC, clinical information, biomarker analysis, etc.) constitute the toolset of MTBs, in order to reach personalized treatment option decisions for a given tumor case. Signalomics represents another technological package providing complementary understanding of the levels of activation of oncogenic signaling pathways in the tumor, along with the sensitivity of these pathways to available treatments. Three pathways—Notch, receptor tyrosine kinase (RTK), and Wnt signaling—are chosen to schematically illustrate some of the oncogenic pathways relevant in breast and gynecological cancers. Created in BioRender. https://BioRender.com/w09e555.

potential targeted therapeutic options for gynecological cancers, including the targeting of epidermal growth factor receptor (EGFR), androgen receptor, poly ADP-ribose polymerase (PARP), and vascular endothelial growth factor (VEGF) (Luque-Bolivar et al, 2020). For instance, in OC, the presence of BRCA mutations leading to homologous recombination deficiency predicts better response rates to PARP inhibitors, either as single agents or in combination with bevacizumab (acting on VEGF) during maintenance therapy (Mirza et al, 2016; Ray-Coquard et al, 2023). Pembrolizumab, a monoclonal antibody against programmed death receptor-1 (PD-1), has shown efficacy in patients

suffering from EC with high mutational burden (Makker et al, 2020).

The advent of personalized targeted therapies represents a transformative moment in the field of oncology, allowing clinicians and researchers to move beyond the "one-size-fits-all" treatments towards more individualized approaches. This paradigm shift is based on a better understanding of the molecular and genetic underpinnings of cancer involving genetic mutations, epigenetic alterations, and tumor microenvironmental factors. Targeted therapies are designed to interfere with specific molecular targets associated with cancer progression. Identification of cancer mutations through whole-genome, whole-exome, or panel gene NGS permits

oncologists to categorize tumors based on their molecular profiles instead of solely morphological descriptions. Modern oncology utilizes a body of knowledge that requires specialized interpretation by experts from multiple fields, including both clinical specialists and basic scientists. In this context, Molecular Tumor Boards (MTBs) have been organized to facilitate the most effective interaction of multidisciplinary specialists (Charo et al, 2022; Kato et al, 2020; Luchini et al, 2020; Rolfo et al, 2018; Tsimberidou et al, 2023).

The purpose of this Perspective is to justify new approaches to tumor phenotyping based on the determination of activities of signaling pathways associated with the tumor, to better

guide the selection of targeted therapies as part of the MTB activities (Fig. 1).

## Molecular tumor boards (MTBs) in cancer management and their current limitations

Despite the accumulating knowledge of tumor genetics and the expanding accessibility of NGS data, implementation of targeted therapies and precision medicine in clinical practice continues to encounter challenges (Lynch et al, 2017). In response to the challenge of NGS data interpretation, numerous medical centers have established MTBs aimed at integrating NGS data with clinical data for informed cancer treatment decisions (Berger et al, 2018). These boards are typically multidisciplinary teams, comprising medical oncologists, surgeons, geneticists, bioinformaticians, molecular biologists, and pathologists, among others (Larson et al, 2021). Their primary objective is to discern suitable treatment choices through integration of genetic analyses, particularly for patients displaying inadequate responses to standard therapies (Bartoletti et al, 2022; Luchini et al, 2020). The resulting recommendations from MTBs are formulated through comprehensive multidisciplinary consultations, taking into consideration key factors such as driver mutations, copy number alterations, structural variations (including gene fusions), microsatellite instability, mutational tumor load, and changes indicative of drug resistance. Deliberations by MTBs yield personalized clinical recommendations, encompassing potential targeted therapies based on the individual tumor's genomic profile, potential enrollment in pertinent clinical trials, or guidance related to the necessity for sequential biopsies (Rodriguez-Rodriguez et al, 2016). The clinical recommendations may include Food and Drug Administration (FDA)-approved, on- or off-label targeted therapies, cytotoxic agents, or radiation therapy, particularly for patients with tumors characterized by DNA repair pathway defects. Studies have demonstrated that patients undergoing MTB-recommended therapy exhibit significantly longer progression-free survival and overall survival compared to patients receiving therapies standardly chosen by physicians (Gremke et al, 2024; Kato et al, 2020; Rodon et al, 2019). Although breast and gynecological cancers are known to

harbor fewer actionable mutations than other solid tumors (Gunderson et al, 2016), several studies have shown improved clinical outcomes for patients receiving MTB-based therapy for gynecological tumors. For example, enrollment of 69 patients with gynecological cancers led to treatment suggestions based on clinical and genomic data for 64 patients (Rodriguez-Rodriguez et al, 2016). Another study highlighted that patients who received MTB-recommended therapies exhibited a 59% clinical benefit rate (Charo et al, 2022). Stratifying clinical benefit rates by Matching Score (MS, calculated by evaluating the number of alterations targeted by the administered drugs divided by the total number of alterations) revealed significant association between higher MS and the clinical benefit rate, with 73% of women with MS ≥40% demonstrating stable disease over ≥6 months, partial response, or complete response (Charo et al, 2022). In contrast, patients with MS <40% exhibited a considerably lower rate of stable disease. A recent retrospective analysis in BC and gynecological tumors demonstrated prolonged progression-free survival in patients who received molecular-matched treatment after the MTB recommendation (Gremke et al, 2024).

The success of precision medicine and the utilization of MTBs in oncology treatment, while substantial, is accompanied by noteworthy limitations (Patel et al, 2018). A common obstacle in translating MTB recommendations into clinical practice lies in disease progression prior to therapy initiation, underscoring the need for early genomic profiling in the disease course (Patel et al, 2018). The emergence of resistance to the chosen targeted therapy, e.g., through activation of alternative signaling pathways, presents another complication and leads to disease progression (Rodon et al, 2019). Further challenges include limited access to drugs or clinical trials, which hinders patients from receiving the recommended regimen, especially in developing countries (Kato et al, 2020; Patel et al, 2018; Rodon et al, 2019). Standardization of clinical approaches is essential for the integration of MTBs into practice to mitigate the heterogeneity in MTB recommendations. Another major difficulty is that driver mutations in many cancers remain unknown; malignant transformations may also be driven by epigenetic changes that alter gene expression profiles, e.g., through

histone modifications, DNA methylation, and miRNA-based alterations (Sharma et al, 2010). Importantly, a simple increase in the gene panel size for NGS diagnostics failed to improve the patient outcomes in MTB practices (Trédan et al, 2025), arguing for the need for additional molecular analysis methods.

To cope with some of these challenges, integration of transcriptomic analysis into precision oncology trials has been made (Rodon et al, 2019; Tsimberidou et al, 2022). For example, in the WINTHER trial, patients were guided to therapy based on either DNA-based NGS or transcriptional analysis that specifically compared tumors to the matched normal tissues (Rodon et al, 2019). Transcriptomic analysis studying all RNA transcripts in a cellular population reveals gene expression profiles using high-throughput technologies such as microarrays and RNA sequencing (RNA-seq) (Cieślik and Chinnaiyan, 2018; Tsimberidou et al, 2023). However, the number of gene expression signatures stably integrated into clinical practice remains limited (Bertucci et al, 2014; Cao et al, 2017; Tsimberidou et al, 2022).

We wish to further elaborate on the fundamental MTB limitation linked to the fact that not all tumors (and not in all aspects) rely on mutational activation/inactivation of crucial cellular mechanisms to trigger or promote oncogenesis and progression. Activation of oncogenic signaling pathways often does not hinge on mutations in genes encoding components of these signaling pathways, but can be achieved through epigenetic alterations in the levels of expression of these components, such as overproduction of activators and/or underproduction of signaling pathway suppressors (Koval and Katanaev, 2018). Even when a deleterious mutation is identified, it is not always pathogenic/cancer-driving, as oncogenic variants are often found in healthy or premalignant tissues, further complicating the interpretations by MTBs (Wahida et al, 2023). Further research is imperative to develop supplementary strategies for tumor analysis, complementing current MTB protocols. The goal of such methods is to rapidly assess the level of activation of oncogenic signaling pathways in the patient's tumor cells, irrespective of the presence or absence of driver mutations. Analytical methods for assessing the levels of activation of oncogenic pathways in tumors need to be

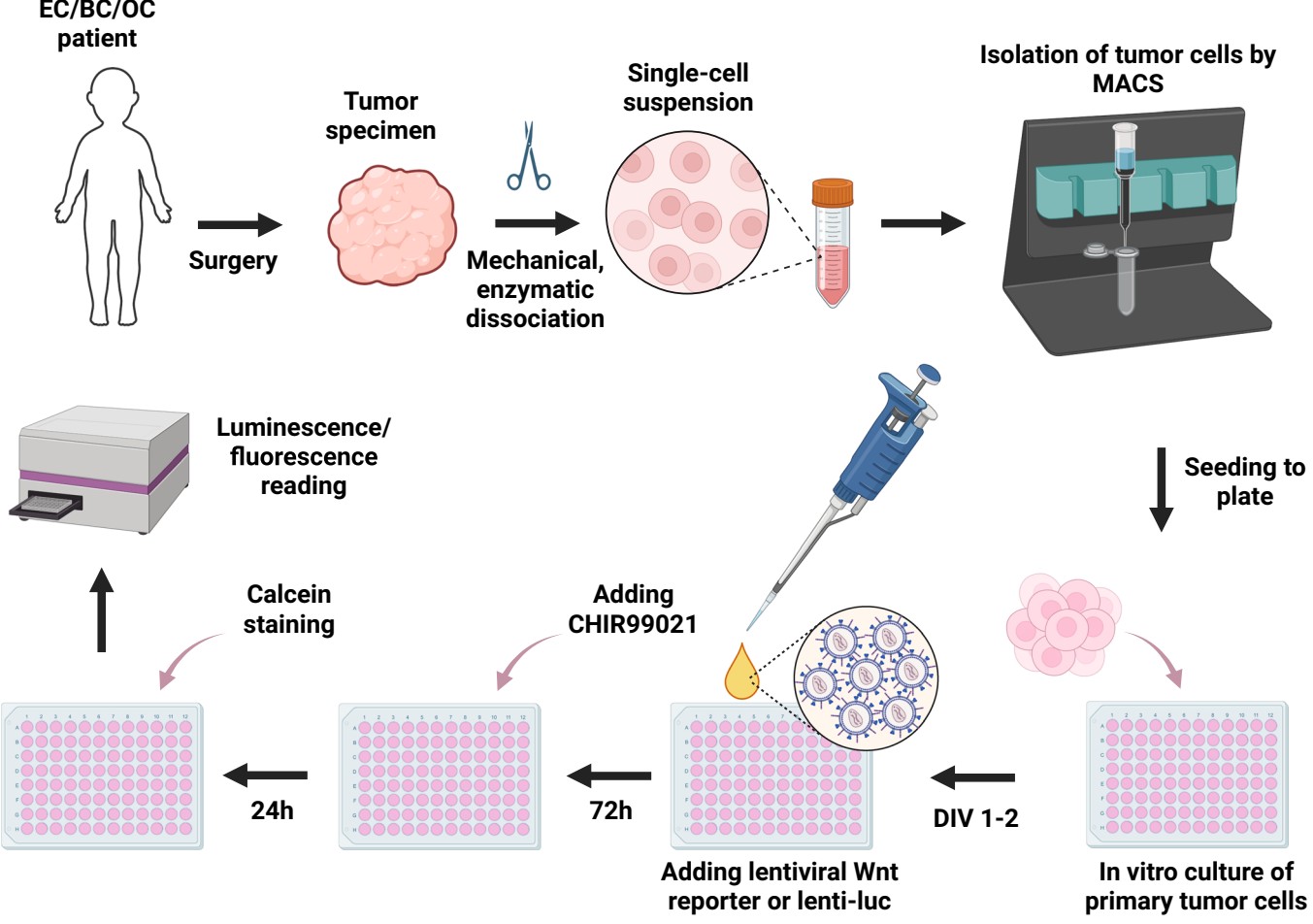

**Figure 2. Workflow of the personalized oncogenic signaling status identification in patient-derived tumor cells.**

EC endometrial cancer, BC breast cancer, OC ovarian cancer, MACS magnetic-activated cell sorting, DIV day in vitro. CHIR99021 is illustrated as the artificial pathway activator for the oncogenic Wnt signaling. Created in BioRender. https://BioRender.com/jqj0y6d.

developed and implemented in the MTB practice. We propose here a new direction called "Signalomics", aimed at studying the activation of signaling pathways associated with the development of the tumorigenic process (Fig. 1). The methodological foundation of this approach is the analysis of signaling pathways activity in primary isolated cancer cells using reporter constructs (Fig. 2). Integrating these insights with existing MTB toolsets will improve outcomes for cancer patients in general, and patients with breast and gynecological cancers in particular, as these types cannot be adequately studied at the molecular level by analysis of genomic alterations alone. Integration of the Signalomics platform into MTB should inform the design of patient-specific treatment plans and more accurate predictions of treatment responses.

## Signalomics as a new MTB unit

We propose Signalomics as a new approach for identifying a repertoire of signaling pathways that are altered or dysregulated in cancer cells. Multiple signaling pathways play important roles in the development and progression of different types of cancer (Fu et al, 2022; Lei et al, 2022; Song et al, 2024; Zeng et al, 2023). In gynecological and breast cancers, the pathways mediated by ER, HER2, the Wnt signaling, and also the Notch, Hedgehog, NFκB, and other signaling pathways represent important oncogenic drivers (Feng et al, 2018; Song et al, 2022) (see Fig. 1 for schematic representation of some of these pathways).

Currently, understanding the status of a particular pathway can be achieved by analyzing the presence of driver mutations by NGS (multigene panel or whole-exome

sequencing), transcriptome analysis, or examining the presence or localization of protein markers, if available, by immuno-histochemistry (IHC) of the patient's tumor tissue (Luchini et al, 2020; Tsimberidou et al, 2023). All of these methods have their limitations. For example, as discussed above, signaling pathways can be activated without mutations (Fattahi et al, 2018; Sharma et al, 2021). Even once a mutation in a signaling component is identified, if it is not a well-established driver mutation, it may have no functional significance for the respective signaling pathway (Sanchez-Vega et al, 2018). Further, DNA sequencing data provide a limited quantitative assessment of the pathway activities. Although transcriptomics offers data on expression levels of multiple target genes of a given pathway, current analysis methods still face difficulties in evaluating the degree of the pathway

activity in a given sample, and even inclusion of tens and hundreds of genes in the activity signature only allows generation of a semi-quantitative score (Drier et al, 2013). Use of different data providers and panels leads to a significant heterogeneity in the interpretation of such data in MTBs (Crimini et al, 2022). IHC methods are largely non-quantitative (although significant progress has been made in tailoring them to pathway activities through, e.g., emerging AI-based approaches (Vorontsov et al, 2024)).

Indeed, the Wnt signaling pathway, as an example, includes about a hundred components and targets (Koval and Katanaev, 2018), and their complex expression changes in individual tumors make it difficult to reach an univocal decision on the pathway activation, especially in quantitative terms. Accumulation of β-catenin in the cytoplasm and its nuclear translocation, a marker of Wnt signaling activation in cancer tissues, can be qualitatively visualized by IHC. However, not all oncogenic signaling pathways have such markers for even a qualitative analysis. Therefore, we are putting forward Signalomics as a new and highly informative platform to complement the MTB practice. Signalomics aims at measuring the levels of activation of a panel of oncogenic signaling pathways in freshly isolated tumor samples. Once the overactivated pathway is identified, Signalomics next permits to rapidly validate the sensitivity of this overactivation to available targeted therapies directed at that particular pathway, providing valuable data for the ultimate decision on the personalized treatment options (Fig. 1).

The Signalomics toolkit for the quantitative examination of the levels of molecular pathway activities includes transcriptional reporter assays (Dai et al, 2012; Shaw et al, 2019). In such assays, a synthesized reporter gene, that upon expression emits colorimetric, fluorescent, or luminescent signals, is placed under the control of relevant promoter-enhancer sequences, which drive the reporter gene expression upon—and in the measure of—activation of the signaling pathway under study. The resulting genetic construct is delivered to cells by viral transduction or non-viral transfection. Lentiviral-mediated reporter gene expression has been successfully employed to detect molecular pathway activities in cell culture and in vivo (Badr and Tannous, 2011; Yde Ohki et al, 2023).

This approach is transferable to primary gynecological cancer and BC specimens to evaluate activation of oncogenic pathways. Primary cells from surgical tumor samples provide a more accurate in vitro tumor model than stable cancer cell lines, as they better retain the cellular functions, growth characteristics, markers, and signaling pathways of the original tissue (Faridi et al, 2018; Piwocka et al, 2024). Short-term cultivation of these primary cells prevents in vitro selection of clones better adapted to artificial conditions (Janik et al, 2016); prolonged culture-driven selection inevitably alters the original tumor cell phenotype (Ince et al, 2015) and should thus be avoided.

Briefly, the protocol to be applied consists of the following steps (Fig. 2). Tumor samples are collected by a certified pathologist after surgery or biopsy and swiftly transferred for subsequent experimentation. The tissue is dissociated both mechanically and enzymatically with the eventual cancer cell population isolation from the bulk by magnetic-activated cell sorting (MACS) or fluorescence-activated cell sorting (FACS). Then, cancer cells are seeded in plates and transduced with lentiviruses carrying reporter construct(s) for the oncogenic signaling pathways under study, such as TopFlash, a standard Wnt/β-catenin-responsive firefly luciferase reporter plasmid (Molenaar et al, 1996; Shaw et al, 2019). If the questioned signaling pathway is active, the transduced cells produce luciferase, and the relative level of pathway activation can be evaluated in each individual specimen by the luciferase assay. To evaluate the transduction efficacy, a control lentivirus is used, expressing luciferase constitutively, meaning its expression is not dependent on the activation of a particular signaling pathway. Finally, to assess the maximum possible level of the Wnt signaling pathway activation in the sample, one group of cells is treated with pathway activators such as CHIR99021 (a GSK3β inhibitor (Ring et al, 2003)), a specific receptor-independent Wnt signaling pathway stimulator. After the treatments, cells are lysed and the luminescence is measured with a spectrophotometer. Overall, the protocol takes <1 week, can be expanded to a panel of signaling pathways and is easily implementable into the routine practice of a general laboratory.

Expanding this Wnt signaling example for a panel of oncogenic pathways, the following parameters will be monitored for each signaling pathway under study:

a) The background (unstimulated) levels of the pathway activation in the tumor. When applied to a panel of pathways, this parameter will provide the molecular data on which pathways—and to which extent—are active in a given tumor. Comparing these data across multiple tumors of the same type will identify the pathway(s) that are unusually active/inactive in the given patient's sample.

b) The maximally achieved levels of the pathway activation in the tumor. To obtain this information, robust pathway activators will be applied. Usage of several activators for each pathway (e.g., CHIR99021 and Wnt3a for the Wnt pathway (Shaw et al, 2019)) will permit accounting for any possible off-target effects of the compounds. The information to be obtained in this manner will be important for two reasons. First, given the inherent heterogeneity of the cancer cells within a tumor, the basal pathway levels will provide a snapshot taken at a given moment of tumor progression to reveal the number of cells currently active in their signaling, while what may drive the tumor progression is rather the potential of this cancer cell population to engage the given pathway at critical time points. In this regard, knowing the maximally achievable oncogenic signaling levels may represent a useful biomarker of tumor progression/aggressiveness. Second, this information will provide clues on the possible cross-talk among different oncogenic pathways in the given tumor, adding to the systems-level understanding of the molecular portrait of the tumor.

c) The levels of pathway suppression upon treatment with available drugs that act as targeted pathway inhibitors. This information will be helpful for the ultimate MTB recommendations.

Application of this approach to the panel of relevant oncogenic signaling pathways will permit rapid capture of the signaling "signature" of the given cancer case. Knowledge of the sensitivity of the overactivated pathways to available targeted therapies will guide the potential treatment regimens. And integration of the Signalomics findings with the other molecular methodologies will empower overall MTB activities (Fig. 1). We envisage that such integration will be

important for the scalability of Signalomics to encompass a large panel of signaling pathways to be assessed simultaneously, promoting the routine integration of this approach to MTB practices.

## Synergy between Signalomics and existing next-generation approaches for MTB decision making

The decision-making process in MTBs is driven by data derived from NGS, encompassing single-cell and spatial transcriptomics, and also from other methodologies, including histopathology of tissues and other omics, such as proteomics and metabolomics (Blume-Jensen and Hunter, 2001; Tsimberidou et al, 2023). These multidimensional lines of analysis, to which we wish to add Signalomics, have the power of providing complex datasets that require data-processing algorithms to integrate and digest the molecular findings, for the sake of individualized patient therapy. The Signalomics data are organically complementary to datasets delivered by the existing MTB technologies, especially NGS. Although MTBs may employ customized workflows developed by local teams (Hamamoto et al, 2022; Lutz et al, 2025), common tools for MTB data analysis have been established (MTBP (Tamborero et al, 2022), Knowledge Connector (Glocker et al, 2020)), incorporating variant calling and gene expression analyses. Signalomics findings will permit higher prioritization of mutations in gene(s) belonging to specific pathway(s). Furthermore, the database obtained upon scaled-up Signalomics implementation will facilitate a systematic enhancement of the interpretation of mutation effects. Similarly, results of gene expression pipelines will be correlated with the Signalomics data to assess the functional effects of overexpressed genes and determine the underlying molecular mechanisms.

We would like to further stress the methodological nexus between Signalomics and transcriptomics. A notable illustration of this synergy is the application of a lentivirus-based library of uniquely barcoded transcriptional reporters for over 40 signal transduction pathways. This approach facilitates concurrent evaluation of the levels of these synthetic reporters with those of NGS (O'Connell et al, 2016). This method can be further advantageous when used with single-cell sequencing, as it permits attribution of pathway levels to specific populations within a sample. Integration of Signalomics and RNA-seq

methods into a single technological package will enhance the value of both and is expected to deliver highly valuable functional datasets for the fast and routine analysis by MTBs.

Proteome analysis approaches, which evaluate posttranslational modifications (PTMs) related to a given pathway activation, particularly phosphorylation, also provide valuable monitoring of some pathway activations. Methodologically, this can be based on high-resolution mass-spectrometry evaluation (Wu et al, 2014), or on reverse-phase protein microarrays (Gahoi et al, 2015). In either case, the quantitative degree of the pathway activity is deduced in such approaches by analysis of multiple PTM sites and their prevalence for multiple proteins within a given signal transduction pathway, overall providing important quantitative information (Thiery and Fahrner, 2024), easily integratable with Signalomics.

Integration of transcriptomic and signalomic data has been demonstrated (O'Connell et al, 2016) to facilitate a more comprehensive understanding of the tumor molecular landscape, offering a holistic perspective on tumor biology. This, in turn, will enable MTBs to complement and rectify complex results obtained from NGS approaches. Consequently, MTBs will be better equipped to make more informed and personalized treatment decisions.

## Conclusions and outlook

Signalomics can include the whole panel of reporters aiming at simultaneous or consecutive measurements of activation levels of a dozen relevant oncogenic signaling pathways, such as the Wnt, RTK, and Notch pathways in breast and gynecological cancers. Developments of this technology will concentrate on the scaling-up of the number of pathways scrutinized, its integration with NGS data, enhancing its sensitivity to minimal tumor quantities, and its robust applicability to fresh-frozen samples. We believe that such an addition to the MTB practice will permit the capture of the molecular features of the tumor cases under study, previously unavailable with the current NGS, transcriptomics, or IHC methods available to MTBs. When complemented with the assessment of signaling effects and growth inhibition upon application of approved or experimental treatments targeting the upregulated oncogenic pathways, Signalomics will further extend its

applicability for the choice of the relevant treatment strategies for the MTB patients.

In addition to the translational significance of Signalomics, it will also offer a large fundamental opportunity to study how widespread activation and co-activation of oncogenic signaling pathways is in individual tumors. By harnessing the power of Signalomics, clinicians can gain deeper insights into the intricate molecular signaling patterns of tumors and leverage this knowledge to inform personalized treatment decisions. Although we focused our Perspective on breast and gynecological cancers, supported by our unpublished proof-of-concept findings, we are confident that Signalomics will be applicable to a wide range of cancers. Overall, we believe that Signalomics will represent a transformative paradigm in precision oncology that bridges the gap between molecular profiling, functional pathway analysis, and individualized treatment. As Signalomics evolves, it will become a routine and informative unit of the MTB workflows, contributing to both a deeper understanding of cancer biology and therapeutic breakthroughs.

## Peer review information

A peer review file is available at https://doi.org/ 10.1038/s44320-025-00125-1

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

## Author contributions

**Tatiana V Denisenko**: Investigation; Methodology; Writing—original draft. **Anna E Ivanova**: Investigation; Methodology; Writing—original draft. **Alexey Koval**: Investigation; Methodology; Writing—original draft; Writing—review and editing. **Denis N Silachev**: Investigation; Visualization; Methodology; Writing—original draft; Writing—review and editing. **Lee Jia**: Conceptualization. **Gennadiy T Sukhikh**: Conceptualization; Supervision. **Vladimir L Katanaev**: Conceptualization; Formal analysis; Supervision; Investigation; Methodology; Writing—original draft; Project administration; Writing—review and editing.

## Disclosure and competing interests statement

The authors declare no competing interests.

