## [Peer Review File · Molecular Systems Biology]

Signalomics for Molecular Tumor Boards and Precision Oncology of Breast and Gynecological Cancers

Tatiana Denisenko, Anna Ivanova, Alexey Koval, Denis Silachev, Lee Jia, Gennadiy Sukhikh, and Vladimir Katanaev

Corresponding author(s): Vladimir Katanaev (vladimir.katanaev@unige.ch)

Review Timeline:

Submission Date:	16th Mar 25
Editorial Decision:	26th Apr 25
Revision Received:	8th May 25
Accepted:	20th May 25

Editor: Jingyi Hou

Transaction Report:

26th Apr 2025

Manuscript Number: MSB-2025-12963

Title: Signalomics for Molecular Tumor Boards and Precision Oncology of Breast and Gynecological Cancers

Author: Tatiana Denisenko

Anna Ivanova

Alexey Koval

Denis Silachev

Lee Jia

Gennadiy Sukhikh

Vladimir Katanaev

Dear Prof Katanaev,

Thank you for submitting your revised manuscript. We have now received feedback from both reviewers. As you will see below, Reviewer #1 - who previously reviewed an earlier version of this manuscript for another journal and is an expert in Wnt signaling in cancer - is now generally satisfied with the revisions. Reviewer #2, who is new to this manuscript and has a background in systems biology, has raised several significant concerns that will need to be carefully addressed.

Without reiterating all the points raised, some of the more critical issues are the following:

- Reviewer #2's Major Point #1 should be addressed to enhance clarity and better tailor the article to a systems biology audience.
- Reviewer #2's Major Point #2 also requires attention. Additionally, the current manuscript exceeds 9,000 words. Please note that for Perspective articles, the recommended limit is 5,000 words and no more than 50 references. The manuscript should be revised accordingly to meet these requirements.

All the other issues raised by the reviewers need to be addressed within reason. As you may already know, our editorial policy allows in principle a single round of major revision, and it is therefore essential to provide responses to the reviewers' comments that are as complete as possible.

On a more editorial level, please address the following:

1. Figures should be removed from the manuscript. The fonts used in the figures are not consistent. Please ensure uniform formatting throughout.
2. The references need to be formatted according to the Molecular Systems Biology reference style. Citations should be listed in alphabetical order and list 10 co-authors of a paper before to add et al.
3. Please provide up to five keywords in the manuscript.
4. Please remove the "Authors' contribution" section, "Availability of data and materials" and "Funding declaration".
5. Please rename "Competing interests" to "DISCLOSURE AND COMPETING INTERESTS STATEMENT"

If you feel you can satisfactorily deal with these points and those listed by the referees, you may wish to submit a revised version of your manuscript. Please attach a covering letter giving details of the way in which you have handled each of the points raised by the referees. A revised manuscript will be once again subject to review and you probably understand that we can give you no guarantee at this stage that the eventual outcome will be favorable.

I look forward to receiving the revised manuscript soon.

Kind regards,
Jingyi

Jingyi Hou, PhD
Senior Editor
Molecular Systems Biology

We realize that it is difficult to revise to a specific deadline. In the interest of protecting the conceptual advance provided by the work, we recommend a revision within 3 months (25th Jul 2025). Please discuss the revision progress ahead of this time with the editor if you require more time to complete the revisions. Use the link below to submit your revision:

*** PLEASE NOTE *** As part of the EMBO Press transparent editorial process initiative (see our Editorial at <https://dx.doi.org/10.1038/msb.2010.72>), Molecular Systems Biology publishes online a Review Process File with each accepted manuscripts. This file will be published in conjunction with your paper and will include the anonymous referee reports, your point-by-point response and all pertinent correspondence relating to the manuscript. If you do NOT want this File to be published, please inform the editorial office at contact@molsystbiol.org within 14 days upon receipt of the present letter.

Reviewer #1:

I have previously reviewed as a report for EMM and it has now been rewritten as perspective to MSB. In this format i think this is a very important contribution to the literature. Going forward increasing only mutational profiling for precision medicine may not increase actionability as many mutation do not lead to predictions for therapeutics. Therefore integrating tumour signalling (signalomics) into the pipeline (and testing this in ex vivo samples) could lead to an improvement in the precision medicine. Obviously there will be a questions in costs of how to deploy this and ensuring this done to a clinical grade.

Overall this is a very written article and is ready for publication.

One minor thing the authors may want to discuss the recent Nature Medicine paper showing a larger genetic panel of mutations does not improve outcomes for patients compared to smaller ones. This adds to the central hypothesis that simply by sequencing in more depth this might not improve Precision medicine and we need consider new tests to add in.

Reviewer #2:

Summary:

In their perspective article, Denisenko, Ivanova, and colleagues recommend the addition of experimental reporter assays to study oncogenic signalling pathways into clinical molecular tumor boards. They provide an overview of current targeted treatments for breast and gynecological cancers, highlight the success of multidisciplinary tumor boards, and discuss ongoing challenges in the field. The authors introduce their approach, termed Signalomics, which aims to measure oncogenic pathway activations in freshly isolated tumor cells.

General Remarks:

The article is written in a well structured accessible style, making it suitable for a broad, interdisciplinary readership. As a perspective piece, it does not present original experimental data but instead offers a conceptual framework. Over the past years, molecular tumour boards have an evolving role in optimizing treatment selection and thus optimising disease diagnosis and prognosis. This perspective complements several recent reviews and perspectives which are advocating for adding new strategies and frameworks into MTBs (e.g. <https://doi.org/10.1038/s41571-023-00824-4>, <https://doi.org/10.3390/cancers15061727>, <https://doi.org/10.1200/PO.20.00508>). Understanding and diagnosing the cancer patient-specific deregulation of cell signalling and its complex interplay is hereby key for further advancing precision oncology. The concept of Signalomics offers an exciting direction for advancing signaling precision oncology. While the concept is promising, in its current form, the article lacks focus, sufficient technical detail, and critical insight into the practical implementation of Signalomics, limiting its potential to meaningfully advance the current discourse.

Major points:

- The authors propose "Signalomics as a new approach for identifying a specific repertoire of oncogenic signaling pathways that are altered or dysregulated in cancer cells." Yet, many important questions and technical details remain unclear:

a) The scalability of the proposed approach is not fully outlined, particularly regarding the "panel of relevant oncogenic signaling pathways." Although the authors cite the 2017 TF-seq study, which profiled transcription factors from over 40 well-characterized pathways, they do not clarify the current or future scale of their proposed framework.

b) Potential challenges of the Signalomics assays are not adequately outlined. There's no mention of how background activity, cross-pathway activation, or off-target effects are controlled for. For instance, the mentioned GSK3 β inhibitor CHIR99021 has known off-target effects [e.g. <https://doi.org/10.1016/j.stemcr.2018.03.023>]

c) While the authors do a good job of outlining the experimental workflow, a brief description of the data analysis/interpretation

and its integration into downstream computational pipelines is needed.

d) Figure 1 suggests that Signalomics can capture cross-talk between signaling pathways, but no explanation is provided on how this would be achieved. The same applies to "Network signalling interactions" where it is unclear whether this refers to the authors' method specifically or to other commonly used methods in signaling pathway analysis.

e) The article refers to unpublished proof-of-concept findings from the authors' own work without providing sufficient detail or evidence. As readers cannot critically evaluate these claims, these references should be removed, outlined in more detail or only given as an outlook in the conclusion section.

- The lengthy, textbook-style sections on Wnt, RTK, and Notch signaling add little to the central argument of integrating Signalomics into MTBs and should be trimmed or removed entirely. Instead, a brief example illustrating how Signalomics assays can inform treatment decisions with e.g. the example of Wnt signalling would better support the case for integrating Signalomics into MTBs. This would also help the authors stay within the recommended length and reference limits for a perspective piece, while allowing more room to elaborate on their core Signalomics concept.

Minor points:

- The paragraph following the statement "there is still no well-established targeted therapy for TNBC and gynecological cancers" requires further context and nuance for TNBC:

a) The poly (ADP-ribose) polymerase (PARP) inhibitors olaparib and talazoparib have received regulatory approval for TNBC patients with germline BRCA1 and/or BRCA2 mutations with additional PARP inhibitors are in advanced stages of clinical development [<https://doi.org/10.1056/NEJMoa2105215>, <https://doi.org/10.1056/NEJMoa1802905>]

b) Immune checkpoint inhibitors have shown clinical benefit in TNBC. The combination of atezolizumab and nab-paclitaxel was approved for PD-L1-positive metastatic TNBC (mTNBC) based on the phase III IMpassion130 trial [<https://doi.org/10.1056/NEJMoa1809615>]. Similarly, pembrolizumab (Keytruda) is approved in combination with chemotherapy as an immunotherapy option for advanced TNBC [<https://doi.org/10.1056/NEJMoa1910549>, <https://doi.org/10.1056/NEJMoa2409932>].

c) Sacituzumab govitecan (SG), a first-in-class antibody-drug conjugate targeting Trop-2 [<https://doi.org/10.1056/NEJMoa2028485>], has been approved for the treatment of metastatic TNBC.

While these therapies may not yet be considered "well-established" in the broadest sense, they represent meaningful targeted approaches and should be acknowledged accordingly.

- Melanoma should be listed as second example for BRAFV600E, as BRAF inhibitors were first approved for melanoma (2011) - nearly a decade before CRC in 2020 [<https://doi.org/10.1056/NEJMoa1103782>].

- The sentence "IHC methods are inherently non-quantitative; etc." appears incomplete and overlooks the existence of (semi-)quantitative immunohistochemistry methods [e.g. <https://doi.org/10.1038/modpathol.2016.176>, <https://doi.org/10.21037/atm-21-2950>]. A more nuanced phrasing could help reflect the current capabilities of IHC more accurately.

- The term "Signalomics" has been used previously in various contexts, both to describe specific frameworks and more broadly as a term for the omics-based study of signaling [e.g. <https://doi.org/10.1016/j.micres.2024.127995>, <https://api.semanticscholar.org/CorpusID:83270920>]. While it's unclear whether this overlap would cause confusion, the authors may wish to consider this for their terminology.

- The 2022 version of GLOBOCAN [<https://doi.org/10.3322/caac.21834>] is now available and could replace the older version for more up-to-date citation, though the numbers and statements have not changed meaningfully

- "Consequently, the insufficiency of available therapeutic strategies has contributed to high mortality rates in patients with these cancers" - is not directly supported from the statements above and could be revised as "insufficiency of available targeted therapeutic strategies" or similar.

- The claim that "integration of the Signalomics platform into MTBs should lead to the discovery of novel treatment options" is unclear/unsupported; a more appropriate phrasing might be "should inform the design of patient-specific treatment plans."

- The reference to Figure 1 in the MTB paragraph is not fitting, as the figure does not depict multidisciplinary teams.

- The authors often use phrases like "various/several studies" but cite only one or two sources. Adding a review or key studies would strengthen these claims.

- The authors briefly mention challenges in interpreting NGS results; providing specific examples could help clarify what issues need to be addressed.

- The statement "Presently, the decision-making process in MTB is predominantly influenced by data derived from NGS, encompassing single-cell and spatial transcriptomics, with less profound involvement of other methodologies including histopathology of tissues and other omics, such as proteomics and metabolomics" overstates the clinical use of single-cell and spatial transcriptomics while undervaluing the continuing importance of histopathology. While it's true that NGS is becoming more central to MTBs and that proteomics/metabolomics are currently underused, the overall claim oversimplifies the current landscape and misrepresents the maturity of several technologies. [<https://doi.org/10.1038/s41587-024-02543-2>]

- The perspective does not reflect on the numerous well-validated computational methods to infer pathway activity and identify oncogenic pathway activation [<https://doi.org/10.1038/s41467-017-02391-6>, <https://doi.org/10.1073/pnas.1219651110>, <https://doi.org/10.1186/s13073-018-0531-8>, <https://doi.org/10.1073/pnas.050658010>]. In this context, the statement "Transcriptome analysis may be challenging to clearly identify the pathways upregulated..." should be adjusted. While it is outside the scope of this perspective, machine learning models trained on known pathway signatures (e.g. foundation and transformer models) should be briefly acknowledged as emerging tools [<https://doi.org/10.1038/s41591-024-03141-0>].

- The recent review "NGS-Guided Precision Oncology in Breast Cancer and Gynecological Tumors-A Retrospective Molecular Tumor Board Analysis" [<https://doi.org/10.3390/cancers16081561>] offers additional relevant background.

Language, presentation & style:

- The perspective is well-structured, with smooth transitions that effectively guide the reader through the article. However, clarity could be improved by presenting key points more concisely and avoiding repetition. Several arguments-such as the importance of early molecular/genomic profiling-are mentioned multiple times rather than being consolidated.
- Several abbreviations (e.g., IHC, HER2) are not defined, reducing accessibility for non-expert readers. For clarity and consistency, "Estrogen receptor (ER) positive" should be revised to "Estrogen receptor positive (ER+)."
- "Fundamental researchers" is not a commonly used term and could be replaced by the more common "basic scientist"
- Gene names and symbols are inconsistently used, especially when listing Wnt signaling targets where a mixture of gene names and symbols is used.
- The sentence "Thus, ER+ tumors are commonly treated with the anti-hormonal therapy, which aims to suppress estrogen signaling, additionally including chemotherapy" could be rephrased for improved clarity and flow-for example: "Thus, ER+ tumors are commonly treated with anti-hormonal therapy to suppress estrogen signaling, often in combination with chemotherapy."
- In the third paragraph of the introduction, the phrase "Compared to these types of BC" should be removed. Subjective terms and unnecessary qualifiers like "quite often" or "invaluable data" should also be omitted or replaced with evidence-based language.
- The authors should revisit the use of definite articles ("the") throughout the manuscript.
- There are multiple grammar errors and typos in the abstract:
 - a) "led to the establishment" instead of "led to establishment"
 - b) "widespread" instead of "wide-spread"
 - c) "multi-specialist" (better: "multidisciplinary")
 - d) "permits" instead of "permit"

Point-by-point responses. Responses are provided in the blue-colored font.

Editorial comments:

a) Additionally, the current manuscript exceeds 9,000 words. Please note that for Perspective articles, the recommended limit is 5,000 words and no more than 50 references. The manuscript should be revised accordingly to meet these requirements.

In revision, significant shortening has been performed, bringing the word count down to 6600 from 9000, and the number of references – to 70 (from 143), despite adding new references and discussions as demanded by the Reviewers.

b) Figures should be removed from the manuscript. The fonts used in the figures are not consistent. Please ensure uniform formatting throughout.

Figures were provided as separate files and were not (and are not in the revision) part of the manuscript file. The fonts in the two Figures have now been unified.

c) The references need to be formatted according to the Molecular Systems Biology reference style. Citations should be listed in alphabetical order and list 10 co-authors of a paper before to add et al.

References are now reformatted to the Molecular Systems Biology reference style.

d) Please provide up to five keywords in the manuscript.

Provided.

e) Please remove the "Authors' contribution" section, "Availability of data and materials" and "Funding declaration".

Removed.

f) Please rename "Competing interests" to "DISCLOSURE AND COMPETING INTERESTS STATEMENT".

Done.

Reviewer 1:

I have previously reviewed as a report for EMM and it has now been rewritten as perspective to MSB. In this format i think this is a very important contribution to the literature. Going forward increasing only mutational profiling for precision medicine may not increase actionability as many mutation do not lead to predictions for therapeutics. Therefore integrating tumour signalling (signalomics) into the pipeline (and testing this in ex vivo samples) could lead to an improvement in the precision medicine. Obviously there

will be a questions in costs of how to deploy this and ensuring this done to a clinical grade. Overall this is a very written article and is ready for publication.

One minor thing the authors may want to discuss the recent Nature Medicine paper showing a larger genetic panel of mutations does not improve outcomes for patients compared to smaller ones. This adds to the central hypothesis that simply by sequencing in more depth this might not improve Precision medicine and we need consider new tests to add in.

We are very thankful to the Reviewer for her/his enthusiastic support of our manuscript throughout the review process. The highly interesting recent publication the Reviewer mentions has now been included into the paper, along with the appropriate brief discussion.

Reviewer 2:

Summary:

In their perspective article, Denisenko, Ivanova, and colleagues recommend the addition of experimental reporter assays to study oncogenic signalling pathways into clinical molecular tumor boards. They provide an overview of current targeted treatments for breast and gynecological cancers, highlight the success of multidisciplinary tumor boards, and discuss ongoing challenges in the field. The authors introduce their approach, termed Signalomics, which aims to measure oncogenic pathway activations in freshly isolated tumor cells.

We sincerely thank the Reviewer for providing these insightful comments, addressing which has significantly enhanced our Perspective article.

General Remarks:

The article is written in a well structured accessible style, making it suitable for a broad, interdisciplinary readership. As a perspective piece, it does not present original experimental data but instead offers a conceptual framework.

Over the past years, molecular tumour boards have an evolving role in optimizing treatment selection and thus optimising disease diagnosis and prognosis. This perspective complements several recent reviews and perspectives which are advocating for adding new strategies and frameworks into MTBs (e.g. <https://doi.org/10.1038/s41571-023-00824-4>, <https://doi.org/10.3390/cancers15061727>, <https://doi.org/10.1200/PO.20.00508>). Understanding and diagnosing the cancer patient-specific deregulation of cell signalling and its complex interplay is hereby key for further advancing precision oncology. The concept of Signalomics offers an exciting direction for advancing signaling precision oncology. While the concept is promising, in its current form, the article lacks focus, sufficient technical detail, and critical insight into the practical implementation of Signalomics, limiting its potential to meaningfully advance the current discourse.

The individual comments / suggestions of the Reviewer are addressed in detail below.

Major points:

- The authors propose "Signalomics as a new approach for identifying a specific repertoire of oncogenic signaling pathways that are altered or dysregulated in cancer cells." Yet, many important questions and technical details remain unclear:

a) The scalability of the proposed approach is not fully outlined, particularly regarding the "panel of relevant oncogenic signaling pathways." Although the authors cite the 2017 TF-seq study, which profiled

transcription factors from over 40 well-characterized pathways, they do not clarify the current or future scale of their proposed framework.

We thank the Reviewer for these comments, which has allowed us to stress (pp. 7-9) how we propose the scalability of the Signalomics to be achieved.

b) Potential challenges of the Signalomics assays are not adequately outlined. There's no mention of how background activity, cross-pathway activation, or off-target effects are controlled for. For instance, the mentioned GSK3 β inhibitor CHIR99021 has known off-target effects [e.g. <https://doi.org/10.1016/j.stemcr.2018.03.023>].

We thank the Reviewer for this important question, which we have addressed in the revision (pp. 6-8).

c) While the authors do a good job of outlining the experimental workflow, a brief description of the data analysis/interpretation and its integration into downstream computational pipelines is needed.

We thank the reviewer for highlighting the importance of data analysis details. We have now modified the section "**Synergy between Signalomics and existing next-generation approaches for MTB decision making**" to include an overview of possible ways how to synergize the Signalomics data with typical existing data analysis outputs, such as variant calling and gene expression (p. 7).

d) Figure 1 suggests that Signalomics can capture cross-talk between signaling pathways, but no explanation is provided on how this would be achieved. The same applies to "Network signalling interactions" where it is unclear whether this refers to the authors' method specifically or to other commonly used methods in signaling pathway analysis.

The cross-talk capturing is now better explained in the section "**Signalomics as a new MTB unit**" (p. 8). Figure 1 is now modified, such that the "Network signaling interactions" line is removed as redundant with the "Crosstalk line", and replaced with the "Sensitivity to available drugs line", to better align the Figure with the overall description now expanded in the "**Signalomics as a new MTB unit**" section.

e) The article refers to unpublished proof-of-concept findings from the authors' own work without providing sufficient detail or evidence. As readers cannot critically evaluate these claims, these references should be removed, outlined in more detail or only given as an outlook in the conclusion section.

As suggested by the Reviewer and also by the Editor, the referral to unpublished findings is removed from the main text of the Perspective.

- The lengthy, textbook-style sections on Wnt, RTK, and Notch signaling add little to the central argument of integrating Signalomics into MTBs and should be trimmed or removed entirely. Instead, a brief example illustrating how Signalomics assays can inform treatment decisions with e.g. the example of Wnt signalling would better support the case for integrating Signalomics into MTBs. This would also help the authors stay within the recommended length and reference limits for a perspective piece, while allowing more room to elaborate on their core Signalomics concept.

Following the suggestion of the Reviewer, and following the Editorial demand to shorten the manuscript, we have removed the sections on Wnt, RTK, and Notch signaling. Expansions on the Signalomics practical aspects have instead been introduced at multiple places.

Minor points:

- The paragraph following the statement "there is still no well-established targeted therapy for TNBC and gynecological cancers" requires further context and nuance for TNBC:

a) The poly (ADP-ribose) polymerase (PARP) inhibitors olaparib and talazoparib have received regulatory approval for TNBC patients with germline BRCA1 and/or BRCA2 mutations with additional PARP inhibitors are in advanced stages of clinical development [<https://doi.org/10.1056/NEJMoa2105215>, <https://doi.org/10.1056/NEJMoa1802905>]

b) Immune checkpoint inhibitors have shown clinical benefit in TNBC. The combination of atezolizumab and nab-paclitaxel was approved for PD-L1-positive metastatic TNBC (mTNBC) based on the phase III IMpassion130 trial [<https://doi.org/10.1056/NEJMoa1809615>]. Similarly, pembrolizumab (Keytruda) is approved in combination with chemotherapy as an immunotherapy option for advanced TNBC [<https://doi.org/10.1056/NEJMoa1910549>, <https://doi.org/10.1056/NEJMoa2409932>].

c) Sacituzumab govitecan (SG), a first-in-class antibody-drug conjugate targeting Trop-2 [<https://doi.org/10.1056/NEJMoa2028485>], has been approved for the treatment of metastatic TNBC. While these therapies may not yet be considered "well-established" in the broadest sense, they represent meaningful targeted approaches and should be acknowledged accordingly.

We have added some of these citations (but we cannot add all due to the strict limit of references) to the relevant place.

- Melanoma should be listed as second example for BRAFV600E, as BRAF inhibitors were first approved for melanoma (2011) -nearly a decade before CRC in 2020 [<https://doi.org/10.1056/NEJMoa1103782>].

Due to the need to strongly reduce the text and citations, we have completely removed the BRAFV600E example.

- The sentence "IHC methods are inherently non-quantitative; etc." appears incomplete and overlooks the existence of (semi-)quantitative immunohistochemistry methods [e.g. <https://doi.org/10.1038/modpathol.2016.176>, <https://doi.org/10.21037/atm-21-2950>]. A more nuanced phrasing could help reflect the current capabilities of IHC more accurately.

We have now nuanced this statement.

- The term "Signalomics" has been used previously in various contexts, both to describe specific frameworks and more broadly as a term for the omics-based study of signaling [e.g. <https://doi.org/10.1016/j.micres.2024.127995>, <https://api.semanticscholar.org/CorpusID:83270920>]. While it's unclear whether this overlap would cause confusion, the authors may wish to consider this for their terminology.

We thank the Reviewer to pointing out these prior examples where the term "signalomics" was introduced. As the term was applied to bacterial communication (first article) and systemic signals in plants (second

article), we do not feel that there may be a confusion regarding the use of the term Signalomics as we introduce it in our Perspective.

- The 2022 version of GLOBOCAN [<https://doi.org/10.3322/caac.21834>] is now available and could replace the older version for more up-to-date citation, though the numbers and statements have not changed meaningfully)

We thank the reviewer for this notion; a new GLOBOCAN article citation now replaces the earlier ones.

- "Consequently, the insufficiency of available therapeutic strategies has contributed to high mortality rates in patients with these cancers" - is not directly supported from the statements above and could be revised as "insufficiency of available targeted therapeutic strategies" or similar.

Now modified as suggested, thank you.

- The claim that "integration of the Signalomics platform into MTBs should lead to the discovery of novel treatment options" is unclear/unsupported; a more appropriate phrasing might be "should inform the design of patient-specific treatment plans."

Now modified as suggested, thank you.

- The reference to Figure 1 in the MTB paragraph is not fitting, as the figure does not depict multidisciplinary teams.

The reference is now removed from this location.

- The authors often use phrases like "various/several studies" but cite only one or two sources. Adding a review or key studies would strengthen these claims.

These statements are now modified in both directions: by removing the word "various" and by adding more citations.

- The authors briefly mention challenges in interpreting NGS results; providing specific examples could help clarify what issues need to be addressed.

Although we are restricted in the word limits, we have expanded on this point in the revised manuscript. As detailed in the updated text of the section "**Signalomics as a new MTB unit**", p.5, significant challenges remain in the quantitative assessment of pathway activity, even with methods that analyze large gene signatures. The semi-quantitative nature of these scores (10.1073/pnas.121965111) and the heterogeneity arising from different data providers and panels within MTBs (10.3390/cancers14133193) illustrate these issues.

- The statement "Presently, the decision-making process in MTB is predominantly influenced by data derived from NGS, encompassing single-cell and spatial transcriptomics, with less profound involvement of other methodologies including histopathology of tissues and other omics, such as proteomics and

metabolomics" overstates the clinical use of single-cell and spatial transcriptomics while undervaluing the continuing importance of histopathology. While it's true that NGS is becoming more central to MTBs and that proteomics/metabolomics are currently underused, the overall claim oversimplifies the current landscape and misrepresents the maturity of several technologies. [<https://doi.org/10.1038/s41587-024-02543-2>]

We have modified this sentence to make it better-tuned to the importance of the other methodologies.

- The perspective does not reflect on the numerous well-validated computational methods to infer pathway activity and identify oncogenic pathway activation [<https://doi.org/10.1038/s41467-017-02391-6>, <https://doi.org/10.1073/pnas.1219651110>, <https://doi.org/10.1186/s13073-018-0531-8>, <https://doi.org/10.1073/pnas.050658010>]. In this context, the statement "Transcriptome analysis may be challenging to clearly identify the pathways upregulated..." should be adjusted. While it is outside the scope of this perspective, machine learning models trained on known pathway signatures (e.g. foundation and transformer models) should be briefly acknowledged as emerging tools [<https://doi.org/10.1038/s41591-024-03141-0>].

Close examination of the references the Reviewer provided, shows that, while they are undoubtedly interesting and valuable for understanding gene expression patterns in cancer biology, these references illustrate the actual problems raised in our perspective. For example, 3 out of 4 (<https://doi.org/10.1038/s41467-017-02391-6>, <https://doi.org/10.1186/s13073-018-0531-8>, and <https://doi.org/10.1073/pnas.050658010>) do not provide direct methods for inferring pathway activity levels from gene expression data, but rather focus on important signatures and classification. The remaining reference (<https://doi.org/10.1073/pnas.1219651110>) does discuss a pathway activity scoring method, but points out that it is semi-quantitative and has several limitations, such as a lack of direct correspondence to established pathways. We have included a discussion of the later reference and other limitations of the NGS approach instead of the statement indicated in your comment in the section "**Signalomics as a new MTB unit**".

We agree with the reviewer that emerging machine learning models, including foundation and transformer models trained on pathway signatures (as exemplified by <https://doi.org/10.1038/s41591-024-03141-0>), represent a promising direction in this field. Despite a word limit, we have included such statement in the same section.

- The recent review "NGS-Guided Precision Oncology in Breast Cancer and Gynecological Tumors-A Retrospective Molecular Tumor Board Analysis" [<https://doi.org/10.3390/cancers16081561>] offers additional relevant background.

We have now included this recent interesting publication into our discussion.

Language, presentation & style:

- The perspective is well-structured, with smooth transitions that effectively guide the reader through the article. However, clarity could be improved by presenting key points more concisely and avoiding repetition. Several arguments-such as the importance of early molecular/genomic profiling-are mentioned multiple times rather than being consolidated.

We have now consolidated these arguments, and overall have made our Perspective more succinct.

- Several abbreviations (e.g., IHC, HER2) are not defined, reducing accessibility for non-expert readers. For clarity and consistency, "Estrogen receptor (ER) positive" should be revised to "Estrogen receptor positive (ER+)."

Corrected, thank you.

- "Fundamental researchers" is not a commonly used term and could be replaced by the more common "basic scientist"

Corrected.

- Gene names and symbols are inconsistently used, especially when listing Wnt signaling targets where a mixture of gene names and symbols is used.

This section is now removed.

- The sentence "Thus, ER+ tumors are commonly treated with the anti-hormonal therapy, which aims to suppress estrogen signaling, additionally including chemotherapy" could be rephrased for improved clarity and flow-for example: "Thus, ER+ tumors are commonly treated with anti-hormonal therapy to suppress estrogen signaling, often in combination with chemotherapy."

Replaced, thank you.

- In the third paragraph of the introduction, the phrase "Compared to these types of BC" should be removed. Subjective terms and unnecessary qualifiers like "quite often" or "invaluable data" should also be omitted or replaced with evidence-based language.

Corrected, thank you.

- The authors should revisit the use of definite articles ("the") throughout the manuscript.

We have now revisited this point throughout the manuscript.

- There are multiple grammar errors and typos in the abstract:

a) "led to the establishment" instead of "led to establishment"

b) "widespread" instead of "wide-spread"

c) "multi-specialist" (better: "multidisciplinary")

d) "permits" instead of "permit"

Corrected, thank you.

20th May 2025

Manuscript number: MSB-2025-12963R

Title: Signalomics for Molecular Tumor Boards and Precision Oncology of Breast and Gynecological Cancers

Dear Prof Katanaev,

Thank you again for sending us your revised manuscript. We are now satisfied with the modifications made and I am pleased to inform you that your paper has been accepted for publication.

Sincerely,
Jingyi

Jingyi Hou, PhD
Senior Editor
Molecular Systems Biology

Reviewer #2:

The authors have addressed all my comments, and I'm satisfied with their revisions. I congratulate them on their work. One last language/style point: the phrase ";etc." at the end of the second-to-last paragraph on page 5 should be removed for better style/clarity.
